# NLOS Identification and Mitigation Using Low-Cost UWB Devices

**DOI:** 10.3390/s19163464

**Published:** 2019-08-08

**Authors:** Valentín Barral, Carlos J. Escudero, José A. García-Naya, Roberto Maneiro-Catoira

**Affiliations:** Universidade da Coruña (University of A Coruña), CITIC Research Center, Campus de Elviña, 15071 A Coruña, Spain

**Keywords:** UWB, machine learning, NLOS identification

## Abstract

Indoor location systems based on ultra-wideband (UWB) technology have become very popular in recent years following the introduction of a number of low-cost devices on the market capable of providing accurate distance measurements. Although promising, UWB devices also suffer from the classic problems found when working in indoor scenarios, especially when there is no a clear line-of-sight (LOS) between the emitter and the receiver, causing the estimation error to increase up to several meters. In this work, machine learning (ML) techniques are employed to analyze several sets of real UWB measurements, captured in different scenarios, to try to identify the measurements facing non-line-of-sight (NLOS) propagation condition. Additionally, an ulterior process is carried out to mitigate the deviation of these measurements from the actual distance value between the devices. The results show that ML techniques are suitable to identify NLOS propagation conditions and also to mitigate the error of the estimates when there is LOS between the emitter and the receiver.

## 1. Introduction

Indoor location systems based on ultra-wideband (UWB) are very popular nowadays due to their ability to provide accurate range estimations based on the signal time of arrival (TOA) or time difference of arrival (TDOA). These systems tend to be more robust and reliable than those based solely on the received signal strength (RSS) [1]. UWB systems exhibit their best performance under line-of-sight (LOS) propagation conditions between the emitter and the receiver. In indoor scenarios, however, this can be difficult to achieve. Obstacles and other interferers can lead to non-line-of-sight (NLOS) situations between the emitter and the receiver, increasing the measurement error. In this scenario, the multipath effect can cause the receiver to detect a signal rebound instead of the direct path. When this happens, the delay between the rebound and the direct path causes a range estimation that is wrongly oversized.

Thus, the detection of NLOS conditions and the mitigation of their effects seems to be a promising way to improve the overall performance of UWB-based systems [2,3,4]. There are several techniques and approaches that follow this idea. Statistical analysis of the resulting range measurements is one of these methods [5,6], whereas another different set of algorithms are based on the study of the channel impulse response (CIR) [7,8,9,10]. The main idea behind these solutions is that the energy of the first path is noticeably greater than the energy of the delayed paths in LOS conditions, whereas this difference tends to shorten in a NLOS scenario. In addition to these methods, NLOS detection can be performed also at a higher logic level, that is, in the location algorithm that uses the range estimation [11]. In this case, some form of additional information, such as a map of the scenario, is needed.

In this work we have followed a different approach. Thus, in order to deal with minimal information on low-cost UWB devices (where the possibility of obtaining the CIR might be limited), only two types of data have been considered: RSS and ranging.

Throughout the article we define the term *measurement* as a pair of estimates resulting from averaging out several raw ranging values (the distance between the emitter and the transmitter) and several raw RSS values obtained at the receiver while it remains in the same position.

The main contribution of this work is an approach to identify and mitigate the effects caused by NLOS propagation conditions. This contribution corresponds to the boxes labeled as *Classification* and *Mitigation* in Figure 1, which shows the typical flow diagram of a location system based on ranging and RSS measurements, where the measurements between multiple anchors (devices placed at known fixed positions) and a target are classified and corrected before being processed by a location algorithm. In our case, as explained in Section 3.2, our features are the averages (μ) of both ranging and RSS. More specifically, this article studies the performance of machine learning (ML) techniques to carry out the classification and mitigation processes in NLOS environments. To focus the study on the feasibility of our proposal, we restrict the algorithms used in the article to classical ML techniques, leaving other methods for future works. To assess the approach in a realistic scenario, the ML algorithms are fed with measurements coming from a measurement campaign employing low-cost UWB devices in indoor environments. From such a set of measurements, some statistics are extracted and used as the features of interest in the classification and regression algorithms. All the measurement data are publicly available in [12] (Please refer to this article if you use these measurements).

The rest of the article is structured as follows. Section 2 explains how the measurement campaign was carried out and how the different LOS and NLOS conditions were achieved. Section 3.1 introduces the ML algorithms considered to analyze the performance of the classifier and the regressive model used for mitigation. The results of these algorithms are presented in Section 4, analyzing the influence of the training method used. Finally, Section 5 presents the conclusions and a projection of future work.

## 2. Measurement Campaign

In this work, we consider a measurement campaign employing UWB devices and carried out in the corridors of the Scientific Area building at the University of A Coruña, Spain [13].

This section is structured as follows: Section 2.1 describes the hardware used as well as the type of data extracted to work with; Section 2.3 describes the environment where the campaign took place, while Section 2.4 summarizes the setup of devices (physical position, configuration, etc.) used in the process; and, finally, Section 2.5 describes the data obtained and some details of interest found during the campaign.

### 2.1. Hardware

For this work we used a set of Pozyx devices [14], a low-cost hardware that integrates a UWB transceiver plus several additional inertial sensors. Because of their price and flexibility (they can be used easily with Arduino or standalone with a computer running the control scripts), they have become very popular in the indoor-localization field. Internally, they use the well known DW1000 chip from Decawave [15]. This chip uses an algorithm called two-way ranging (TWR) [16] to estimate the distance between the emitter and the receiver. In this algorithm, firstly, the initiator sends a poll signal to the responder and records the timestamp of this transmission. The responder receives the signal and records the timestamp at the very first moment it detects the incoming signal. After that, the responder sends the response to the initiator adding to the message the timestamp of the moment just before sending the signal. Finally, the initiator can use its own timestamps (the one at the moment of emission and the one at the moment of reception) plus the timestamps of the responder (the one marking the moment of reception and the one just after sending the answer) to calculate the time of flight (TOF) between the two devices. After this, the distance is obtained multiplying this value by the speed of light.

To detect the main path (i.e., the one with shortest distance between the emitter and the receiver), the Decawave DW1000 uses a leading edge detection (LED) technique that relies in a predefined threshold to detect this first path [17]. The quality and accuracy of this detection can be affected by the physical propagation characteristics of the scenario and the actual location of the devices. Thus, in NLOS conditions, a secondary delayed path can be wrongly selected instead of the main one when its energy does not exceed the corresponding threshold value. Additionally, there are some other problems that can affect the overall accuracy of the ranging estimations, such as noise or bias [18] depending on the distance between the devices.

Pozyx devices can output several pieces of information besides the range estimation, including an estimation of the received RSS (possibly biased [18]), signal parameters such as the pulse repetition frequency (PRF) or the detection threshold value, and even a set of samples of the CIR. However, although it is based on the DW1000 chip, Pozyx hides behind its application programming interface (API) some of its internal values, such as certain power estimations or some data from the accumulator.

As in [7,8,9,10], we could use the provided CIR, but in the case of the Pozyx hardware there are several restrictions that must be taken into account:The CIR must be extracted from the chip sequentially in chunks of data through the serial port, which requires to download 4064 bytes per CIR estimate [19]. That is, the latency introduced to obtain this parameter is about 300 ms for each CIR estimate.CIR measurements correspond to the wireless channel between the the emitter and the receiver at a given time instant. However, these two devices could be different from the ones involved in the ranging process depending on the mode of operation considered. For example, Pozyx devices allow for a so-called *remote* mode in which a node *A* (typically connected to a computer to receive the measurements) can command a remote node *B* to perform a ranging operation between the node *B* itself and a third node *C*. In this case, the CIR available at *A* corresponds to the channel between *A* and *B*, instead of the channel between the two nodes involved in the ranging process, i.e., *B* and *C*. Thus, NLOS detection would be possible only between nodes *A* and *B*, rather than between nodes *B* and *C*, which are the ones that really participate in the ranging process.

With the abovementioned constraints, and if the goal is to work with a low-cost UWB solutions (e.g., the one from Pozyx) in real-time applications where the use of CIR has some limitations, the analysis of the CIR measurements to detect NLOS conditions is a technique that may not be approachable under certain restrictions. Alternatively, Pozyx devices provide RSS and range estimates very fast, up to 100 different measurements per second, which can feed real-time positioning algorithms.

### 2.2. LOS Versus NLOS

In this work, we identify three different scenarios of interest as in [6,20]: LOS, NLOS *Soft*, and NLOS *Hard*:LOS scenario. In this case, both emitter and receiver have no obstacles between them and their separation distance is enough to ensure good communication between them.NLOS *Soft* scenario. Here there is an obstacle that obstructs a possible LOS, so that the main signal path has to go through it (e.g., a wall). In this case, the main and secondary paths reach the receiver attenuated by this obstacle, causing that the RSS do not correspond to the distance, as in the LOS case.NLOS *Hard* scenario. In this case, the emitter and the receiver are physically located in such a way that the secondary paths are received with more RSS than the main one. Basically, the obstacles ensure that the main path is considerably attenuated, or even completely blocked, whereas the reflected paths easily reach the receiver. Thus, in this NLOS *Hard* scenario, the receiver will always intercept a secondary path, which is a delayed version of the main path.

### 2.3. Environment

A measurement campaign was performed in one of the corridors of the Scientific Area building, located in the Elviña Campus, at the University of A Coruña, Spain. To mimic the conditions of interest (LOS, NLOS *Hard* and NLOS *Soft*), a specific zone of the building was chosen. This zone is in the middle of two corridors and has a contiguous room, as shown in Figure 2. Thus, the three propagation conditions are obtained by placing the receivers at the points *A*, *B*, or *C*, corresponding respectively to NLOS *Hard*, LOS, and NLOS *Soft*. Hence, with the anchor in the position *A*, the ranging estimation is always overestimated, since the direct path is totally blocked and only a signal rebound can reach the receiver. With the anchor in the position *C*, the signal corresponding to the shortest path reaches the receiver, but attenuated. Finally, when the anchor is placed at *B*, clear LOS propagation conditions are guaranteed between the emitter and the receiver. Notice that in indoor scenarios such as the one considered for the measurements, there is always multipath, even when LOS is guaranteed.

### 2.4. Hardware Setup

To record the measurements, three Pozyx devices are placed in the scenario as the reference anchors at points *A*, *B*, and *C* (see Figure 2a) to record measurements under the three different propagation conditions considered in this work, i.e., NLOS *Hard*, LOS, and NLOS *Soft*, respectively. More specifically, the first anchor is placed at point *A*, in an adjacent corridor without LOS (NLOS *Hard* condition). The second one is located at point *B*, thus allowing for a good LOS propagation condition. Finally, the last one is placed at point *C*, inside a room adjacent to the main corridor (NLOS *Soft* condition), but isolated by a wall.

To perform the ranging measurements, a target device, that we refer to as the *tag*, is placed in the scenario, along the corridor (see Figure 2a), at different distances with respect to the anchors. To speed up the measurement process, instead of a single tag, we measure with a set of five tags mounted on a tripod (see Figure 2b) with a separation of 20 cm between them.

Besides the tags and the anchors, an additional Pozyx device placed at point *M* (see Figure 2a) is also employed in the measurements. This device is attached to a computer and its mission is to initiate a remote ranging process between the anchors and the tags placed on the tripod. Thus, the process starts with a request from the device *M* to the first anchor (acting as an UWB emitter) to perform a ranging measurement between such an anchor and the first tag (acting as an UWB receiver) in the tripod. The resulting measurement is then returned to the device *M* and is recorded on the computer. This process continues with the remaining tags on the tripod and, next, with the remaining two anchors (see Figure 2a). The tripod remains static for 5 min at each position, repeating the process in such a way that multiple measurements are obtained for each tag position. After this time, it is moved 1 m away from its location. That is, 5 new tag positions are obtained with a spacing of 20 cm between them. In total, the tripod is moved between 3 m to 15 m in a straight line from the anchor *B* (see Figure 2a). The measurement records are tagged with a timestamp and the corresponding actual distance between the anchor and the tag.

### 2.5. Measurements Analysis

Although the measurement campaign was performed varying the distance between the anchor *B* and the tag from 3 m to 16 m (actually 15.8 m), not all recorded measurements were finally included in the set used to train and test the algorithms. Figure 3 shows the range measurements provide by the Pozyx hardware at different distances between tags and its respectively anchors, *A*, *B*, and *C*, respectively. It can be seen that for distance values starting around 9 m, the Pozyx devices produce erroneous ranging values. In particular, the same ranging and RSS values are repeated, regardless of the actual distance between the tags and the anchors. This is a bug that has been reported to the Pozyx company and they are working in a future firmware update to solve this issue. For this reason, only the measurement data up to 9.5 m are considered, discarding the rest.

Figure 3 also shows a noticeable difference between LOS and NLOS *Hard* measurements. Whereas LOS measurements are fitted to a linear function with unit slope (hence providing good ranging estimates), NLOS *Hard* measurements have a different slope and, in addition, exhibit a much higher variance, following a random behaviour (depending on the environment), and leading to a range overestimation. On the other hand, NLOS *Soft* measurements show a similar behavior than LOS ones in positions close to the anchor (i.e., short distance values). However, from a certain distance (about 8 m), its behavior becomes more random due to a considerable drop in the RSS for all the signal paths, and this multipath propagation causes errors in the distance estimation.

Figure 4 shows the RSS value versus the distance between an anchor and a tag for the three propagation conditions: LOS, NLOS *Hard*, and NLOS *Soft*. The RSS values corresponding to the LOS case are higher than that of both NLOS cases. In the NLOS *Hard* case, this is because the path followed by the signal is always longer than the direct path, hence experiencing a stronger attenuation. In the NLOS *Soft* case, the obstacle obstructing the LOS causes an additional attenuation.

## 3. Machine Learning

This section presents the ML techniques considered for *classification*, i.e., to classify LOS and NLOS (*Hard* and *Soft*) propagation conditions based on ranging and RSS estimates provided by low-cost UWB hardware. ML techniques are also employed, in a subsequent step, for *mitigation*, i.e., to compensate for the errors obtained in the ranging estimates. Notice that, although some methods are used in both cases, their configuration and utilization are independent between them.

This section is structured as follows. Section 3.1 describes each considered algorithm and its characteristics, whereas Section 3.3 describes the method used to find the best hyperparameters for each case. Section 3.2 describes the statistics (features) selected to feed the algorithms. Finally, Section 3.4 discusses the issues derived from having a discrete set of measurement points and the different approaches used to reduce their impact in the final results.

### 3.1. Algorithms

In this section we present classic ML algorithms applied for classification and/or mitigation. The performance of this algorithms is assessed by means of different Matlab^™^ (The MathWorks, Inc., Natick, MA, USA) functions included in the Statistics and Machine Learning Toolbox. In the particular case of the Gaussian process (GP), the GPML library was used [21]. Additionally, due to the multiple combinations of execution and the computational cost, the algorithms are compiled and packed to run in the servers of the Fundación Pública Galega Centro Tecnolóxico de Supercomputación de Galicia (CESGA) [22], a supercomputing center located in Santiago de Compostela, A Coruña, Spain.

#### 3.1.1. Binary Decision Tree

The binary decision tree algorithm [23] is a simple learning model that tries to map some input variables into a target one. It works splitting the input variables into different branches according to the value of some features until reaching the leaves. These leaves represent some values of the target variable. The binary decision tree algorithm can be used for classification (classification tree) and regression (regression tree). In the first case, the output variables are discrete (the different classes), whereas in the second case they can get any value in some interval. Classification and regression trees are simple and easy to interpret, but in some scenarios do not generalize well (tends to overfit). In this work, binary decision trees are used for both classification and mitigation.

#### 3.1.2. Support Vector Machine

Support vector machine (SVM) is a classic supervised ML algorithm originally described in [24] and used for both classification and regression problems. The main idea behind this algorithm consists in finding the hyperplane that maximizes the distance between the classes or values of interest. To perform its work, SVM relies on the concept of *kernel* function. A *kernel* is a function that can transform a low-dimensional space into a higher-dimensional one, hence non-separable problems can be converted into separable ones. There are many kernel functions (linear, Gaussian, Polynomial, etc.) and another set of hyperparameters that must be selected for each instance of an SVM algorithm. As with other ML algorithms, this can be achieved using Bayesian optimization (as described in Section 3.3).

SVM can be used for binary classification, even though it can be also considered for multi-class classification problems when using the so-called “one versus all” strategy [25]. In addition, it can be applied to regression problems with slight modifications [26]. In this work, SVM is considered both for classification and mitigation.

#### 3.1.3. k-Nearest Neighbors

k-nearest neighbors (k-NN) is an algorithm employed in classification and in regression problems [27]. It is based on grouping features according to a given distance metric. There are different metrics that can be used in a k-NN algorithm: Euclidean, Mahalanobis, City block, Minkowski, cosine, etc. Besides the metric considered, another important configuration parameter is the number of neighbors taken into consideration.

#### 3.1.4. Gaussian Process Regression and Classification Models

A GP is a generalization of the Gaussian probability distribution in which the distribution does not describe a scalar random variable, but the properties of a function. Based on this idea, it is possible to build regression and classification models with high accuracy and performance [28].

#### 3.1.5. Generalized Linear Models

Generalized linear models (GLMs) [29] are a special variant of nonlinear models that use approximations with linear methods. Among the different configuration parameters, one of the most important is the response distribution type. If, in the case of a linear model, this distribution is assumed to be normal with a mean µ, in a GLM it can follow other functions, such as binomial, Poisson, Gamma, or inverse Gaussian.

### 3.2. Input Features

Since we are considering low-cost UWB devices, statistic parameters computed from RSS and ranging estimates are considered as input features to the ML algorithms instead of CIR estimates. To determine the most suitable combination of statistical parameters we rely on an authors’ previous work [30], which concludes that the most appropriate features to be selected to train and test ML algorithms are the mean and the standard deviation of both ranging and RSS.

### 3.3. Bayesian Optimization

Bayesian optimization is an optimization technique considered for the so-called *black-box* functions [31]. It is a search strategy that tries to find the optimal values of a function in situations where the evaluation of the function is very expensive, especially in terms of time. Bayesian optimization provides a more efficient search strategy than grid or random search. It is based on the use of a surrogate model of the objective function *f*. Typical surrogate models for Bayesian optimization are Gaussian processes. The model uses prior knowledge and previous observations of *f* to generate a posterior estimation of the objective function. Finally, an acquisition function is used on this estimation to propose a new sampling point. This is an iterative method. In this work, the Bayesian optimization approach was employed to find the best suitable hyperparameters of the considered ML algorithms. For the sake of reproducibility and repeatability of the results, and taking into account that in our case each configuration (jumping factor) and each algorithm yields a different set of parameters, we have included these parameters attached to the manuscript according to the Data in the MDPI system.

### 3.4. Discrete Measurement Points

Due to the discrete nature of the measurement points, i.e., the positions of the tags during the measurement campaign (see Figure 2a), the algorithms considered for classification and mitigation can output different results depending on the set of measurements employed for training and testing. If we use the same measurement points in the training and test stages, they could overfit the results on the basis of these positions. However, in a realistic situation, not all possible tag positions can be measured beforehand to be considered as training data, yielding a performance loss compared to the ideal situation in which all distances in the test have been also considered in the training. To assess the impact of this situation, the experiments in Section 4 employ a strategy to exclude from the training a certain amount of measuring points considered in the test.

We define a strategy to split the set of all measurement points into two subsets named A and B. First, we define a method based on a so-called *jumping* parameter, *j*, to guarantee that A and B are disjoint sets. The distance value $d_{n}$ (expressed in meters) between the n-th measurement point and the anchor *B* is defined as
(1)$$d_{n}=3+0.2\left(j+1\right)n,\;n=0,1,2,\cdots ,$$
where 3 is the distance value between the anchor *B* and the first measurement point (which corresponds to n=0), 0.2 is the spacing between consecutive measurements points, and *j* is the aforementioned *jumping* parameter, which represents the number of measurement points to be skipped when selecting the measurement points to be included in a set. Note that if j=0, then all measuring points are selected.

Therefore, the set A is defined by setting *j* to a specific value. For example, if j=1, then one out of two measurement points are skipped, i.e., the measurement points corresponding to the distance values $d_{n}=3,3.4,3.8,4.2,\cdots$ are considered. In fact, the impact of the value of *j* on the performance of the ML algorithms is also evaluated in this work.

Once the sets A and B are defined based on a given value of j>0 according to Equation (Equation 1), we consider their data as the input for the training and test sets. More specifically, the elements of the training set are selected randomly from A, whereas the test set consists of the remaining elements of A, not included in the training set, plus a number of elements randomly selected from B (respecting the proportion of elements taken from A). Notice that if j=0, then the elements of both training and test sets are randomly selected from the set of all measurement points. In addition, note that, although randomly chosen, the training set is constructed ensuring that it includes measurements from all the points in A.

## 4. Results

This section summarizes the results of the considered experiments. More specifically, the classification results are detailed in Section 4.1, whereas those corresponding to the mitigation are presented in Section 4.2.

### 4.1. Classification

Figure 5 shows the performance of a binary classifier trying to identify LOS and NLOS (including both *Soft* and *Hard*); and a ternary classifier to distinguish among LOS, NLOS *Soft*, and NLOS *Hard* conditions. In this experiment, a *jumping* factor j=1 is set according to Equation (Equation 1).

The figure of merit considered in the results is the $F_{1}$-score, which is a widely used indicator to evaluate the classifier’s performance because it takes into account the precision (measurement of false positives) and recall (measurement of false negatives) in the classification to compute the final score:(2)$$F_{1}=2\;\cdot \;\frac{\mathrm{precision}\;\cdot \;\mathrm{recall}}{\mathrm{precision}+\mathrm{recall}}.$$
Different methods can be used to calculate the $F_{1}$-score with more than two classes [32], being the macro average technique the one selected in this work.

According to the results shown in Figure 5, the binary classifier obtains the best results, with $F_{1}$-score values exceeding 0.9 for all the considered ML algorithms. However, when using the ternary classifier, distinguishing between the NLOS *Soft* and the NLOS *Hard* cases yields a significant performance degradation since such a distinction is difficult.

With respect to the considered ML algorithms, the best results in Figure 5 corresponding to the ternary classifier are achieved by the k-NN and the GP algorithms. However, when we employ the GLM algorithm, we observe how it does not get such good results in the ternary classification. The results obtained seem to be consistent with those available in other works such as [33], with similar values of $F_{1}\approx 0.9$ for binary Trees when two classes are considered. Figure 6 shows the $F_{1}$-score values of the algorithms with respect the jumping factor *j* according to Equation (Equation 1). Recall that increasing the value of *j* leads to a larger separation between the measuring points of the training set, hence including new points in the test set. The larger the distance between training measurement points, the more difficult is for the algorithms to classify because the number of new measuring points during the test and the separation from the training points is larger. Obviously, we can see how the best results are achieved for j=0.

Two curves are shown in Figure 6 for each considered ML algorithm: one for the binary classifier (continuous line) and one for the ternary (dashed line). Again, in all cases, the performance of the ternary classifier is worse than that of the binary classifier for all *j* values. More specifically, for j≥1, all binary classifiers work better than the best ternary classifier. Regarding the algorithms, the k-NN, the binary Tree and the GP exhibit the best performance, especially for the ternary classification, producing a solid estimation regardless of *j* and the classification scenario (two or three classes). Finally, the SVM and the GLM exhibit a higher error rate for the ternary classification, whereas they produce values similar to those obtained when the classification is limited to two classes.

### 4.2. Mitigation

Figure 7 shows the mean absolute error (MAE), expressed in meters, of the mitigation process versus the jumping factor, *j*, for the considered algorithms described in Section 3.1. The 95 % confidence intervals are also plotted in the figure as error bars around the MAE. Figure 7 shows the ability of each algorithm to correct the deviations of the measured ranging values with respect to the actual distance value between the anchor and the tag. In Figure 7 we can see the results of the algorithms for each condition, i.e., LOS (Figure 7a), NLOS *Soft* (Figure 7b), and NLOS *Hard* (Figure 7c). In all the graphs in Figure 7, the dashed red line (labeled as “Raw”) corresponds to the MAE of the raw measured ranging value, without mitigation. We can see in Figure 7a that, in the LOS scenario, only GLM and GP are able to reduce the raw MAE consistently, regardless of the distance between the training measurement points. However, when using the Tree and SVM algorithms and for increasing values of of *j*, the mitigation does not improve the raw MAE, but actually makes it worse.

Figure 7b shows the mitigation results after applying the ML algorithms to the measurements corresponding to the NLOS *Soft* scenario. As in the previous case for LOS, the mitigation based on GLM and GP is the one with the best results, especially the GP, whose MAE is reduced in a very solid way. On the other hand, the Tree algorithm obtains the worst results and is unable to reduce the raw MAE for j>1. SVM barely reduces the raw MAE for j<4, but it degrades for j≥4.

The results corresponding to the NLOS *Hard* measurements are shown in Figure 7c. In this case, the MAE reduction is very significant: a raw MAE of about 1.9 m is reduced drastically to become lower than 0.2 m. One of the reasons for this result is the enormous variance of the measurements under NLOS *Hard* conditions. Hence, abnormally erroneous measurements have a big impact on the final value of the MAE. The mitigation algorithms homogenize the data and, in the absence of overfitting, the output estimations follow a smooth distribution. However, notice that, to achieve these results, it is necessary to train the algorithms with measurements coming from an NLOS *Hard* scenario, something that can be difficult to achieve in a real situation. In fact, the training is totally dependent on each specific environment and its propagation conditions since the ranging measurements have different bias and error levels depending on the shape of the rooms, obstacles in the line of sight, building materials, etc. In summary, the mitigation in the NLOS *Hard* scenario cannot be easily generalized.

Figure 8 shows the results for LOS (Figure 8a) and NLOS (Figure 8b) when only two classes are taken into consideration. In this case, the NLOS set includes both NLOS *Hard* and NLOS *Soft* measurements. As expected, the results shown in Figure 8a for the LOS mitigation are almost identical to those shown in Figure 7a for the ternary classification. There are only small variations due to the random realizations of the experiments, but the overall performance of each algorithm is the same. Again, mitigation based on GLM and GP can successfully reduce the raw MAE, whereas SVM, and especially the Tree algorithm, fail in this task.

For the NLOS scenario, however, the results shown in Figure 8b are different. The original error, in this case, is about 1 m since the MAE is now calculated using the error of the NLOS *Soft* and NLOS *Hard* measurements. The mitigation based on the GP algorithm produces the best results, but this time the reduction in the MAE is smaller than in the previous experiment when NLOS *Soft* and NLOS *Hard* were mitigated separately. This can be explained by the fact that the measurements of both scenarios are quite different in terms of ranging and RSS, and consequently, the mitigation algorithms have to integrate extremely heterogeneous data. Thus, the final error values are going to be larger with the single NLOS set than in the version in which NLOS *Hard* and NLOS *Soft* are mitigated independently. The rest of the algorithms achieve also a poorer result compared to the ternary classification and, even though all of them obtain a MAE reduction with respect to the raw MAE, they remain far away from the values reached for the ternary classification.

As in the case with the three classes, the mitigation of the NLOS measurements requires a set of measurements under such a condition, something that is very dependent on the specific environment where the measurements are recorded. Thus, in a real scenario, without a prior characterization (i.e., measurement campaign), it seems more feasible to mitigate only the LOS measurements, whereas the rest (any measurement not classified as LOS) are simply ignored.

## 5. Conclusions and Future Work

In this paper we analyzed the performance of ML techniques applied to classification and error mitigation in low-cost UWB systems when NLOS propagation conditions are present. We relied exclusively on ranging and RSS estimates provided by low-cost UWB devices. Additionally, we considered that NLOS propagation can be split into two different conditions: NLOS *Soft* and NLOS *Hard*, depending on the type of obstacles that affect the signal. The features of a binary classifier (LOS versus NLOS) and a ternary classifier (LOS versus NLOS *Soft* versus NLOS *Hard*) were obtained, and the results of different ML algorithms (k-NN, Tree, GLM, SVM, and GP) were compared for this two cases. Such algorithms were also employed to mitigate the error of the ranging estimates provided by the hardware. Again, the mitigation performance was tested on the three LOS, NLOS *Soft* and NLOS *Hard* conditions.

All the results are based on experimental data obtained from measurement campaigns carried out by the authors and are available to the scientific community. The results revealed that ML techniques are suitable for binary and ternary classification, obtaining significant better results in the first situation when only LOS and NLOS are considered. The k-NN and the GP algorithms showed the best behaviour in this phase. However, applying ML techniques to mitigation seems to be a much more challenging problem. In view of the results, mitigating estimations obtained under LOS propagation conditions yields a small improvement in terms of MAE. However, when NLOS propagation arises, although the performance of the mitigation is great, it is very dependent on the particular scenario where the measurements were captured.

Future work includes performance analysis of more complex classification and mitigation algorithms (such as neural networks) as well as testing of all methods when applied to other public measurement sets, as for example the one in [34]. In addition, as a final test to validate the approach proposed in this paper in a real environment, a classification and mitigation test will be performed in a scenario different than the one considered to extract the measurements for the training set of the different algorithms. Thus, it will be analyzed if the method described in this work is general enough to be used, without modifications, in other distinct places. Finally, there will also be a study of the impact on the positioning produced by the detection (alone or followed by a mitigation stage) of adverse NLOS effects in the UWB measurements that feed the location algorithms.

## Figures and Tables

**Figure 1 sensors-19-03464-f001:**
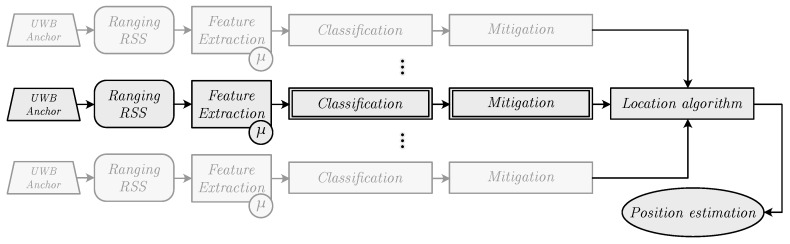
Ranging-based location system block diagram.

**Figure 2 sensors-19-03464-f002:**
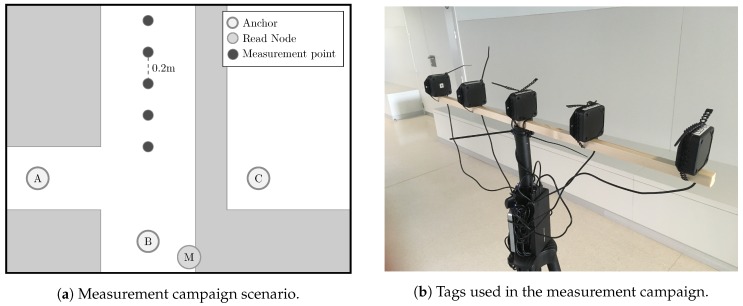
Measurement campaign elements.

**Figure 3 sensors-19-03464-f003:**
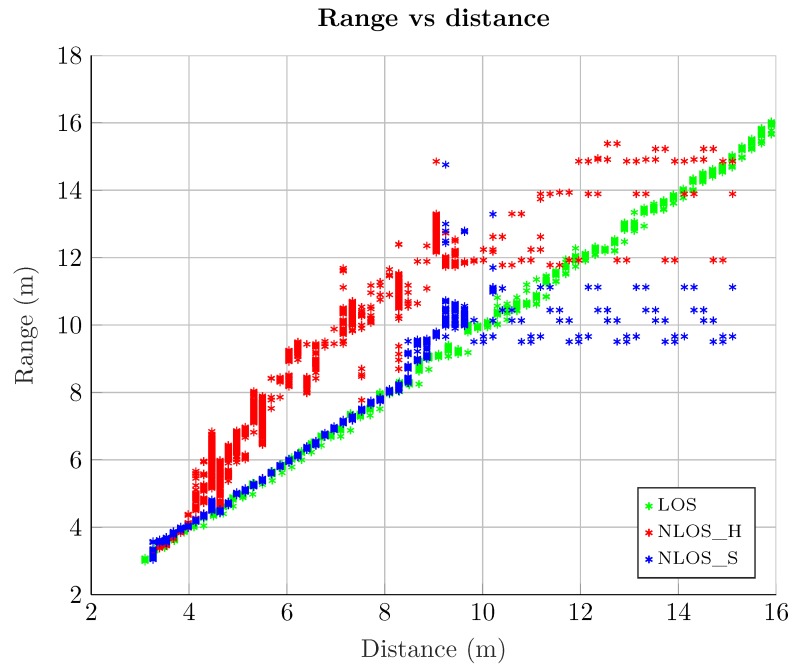
Raw measurements. Estimated range vs. actual distance.

**Figure 4 sensors-19-03464-f004:**
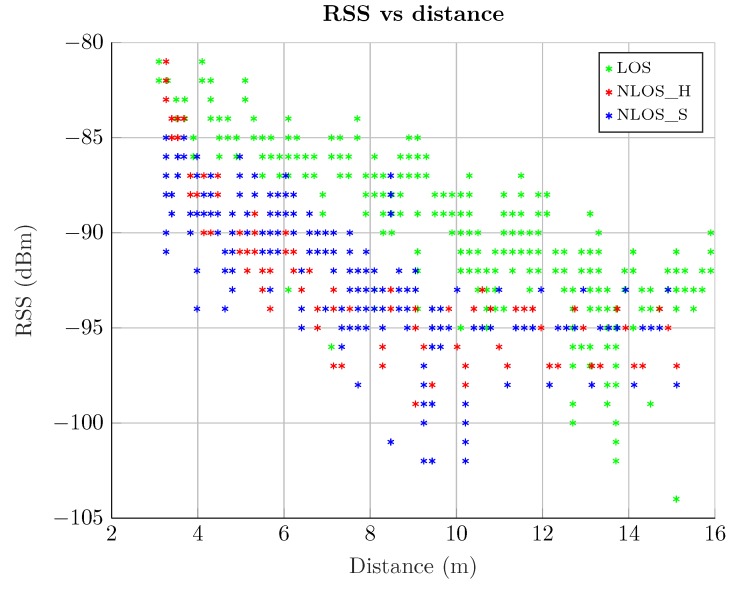
Raw measurements. RSS versus actual distance.

**Figure 5 sensors-19-03464-f005:**
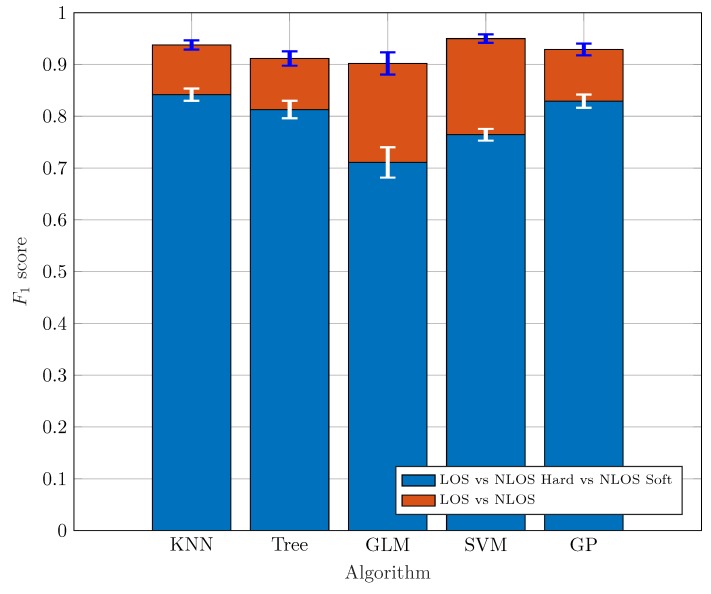
$F_{1}$-score for jumping factor j=1.

**Figure 6 sensors-19-03464-f006:**
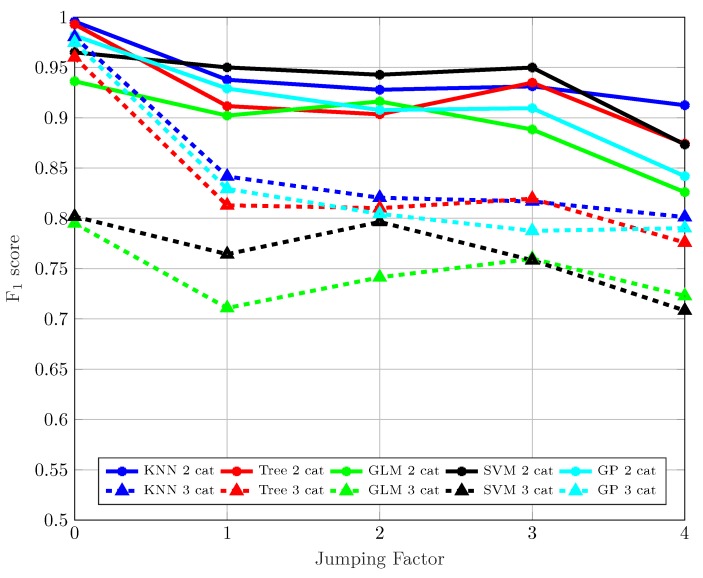
$F_{1}$-score vs. jumping factor, *j*.

**Figure 7 sensors-19-03464-f007:**
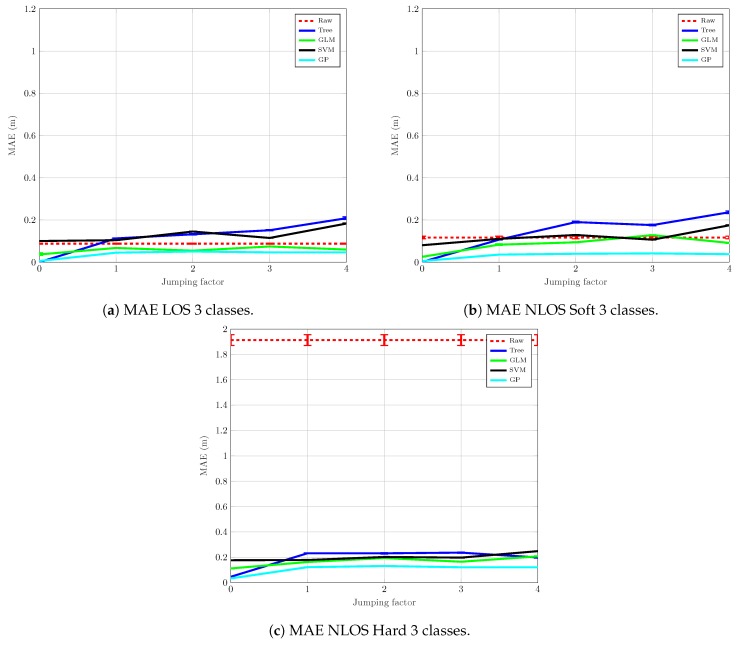
Mitigation MAE with 3 classes.

**Figure 8 sensors-19-03464-f008:**
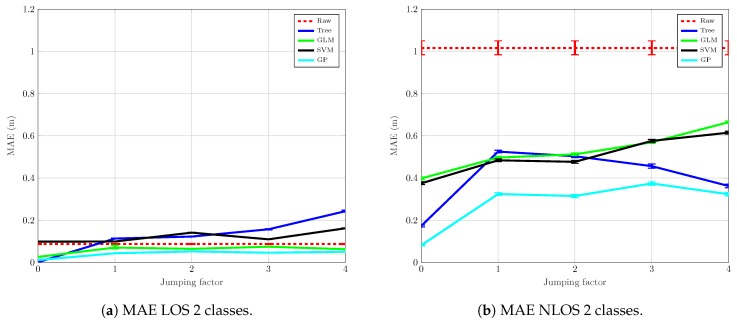
Mitigation MAE with 2 classes.

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
