# Peer review of "NLOS Identification and Mitigation Using Low-Cost UWB Devices"

_sensors, 2019, doi:10.3390/s19163464_

Round 1

Reviewer 1 Report

The authors propose an interesting low-complexity approach for detecting NLOS and improving ranging accuracy for indoor localisation applications using UWB devices. The availability of the captured datasets is a great addition. However, a number improvements / clarifications need to be made.

The related work section should be extended and compared with other SoA approaches for UWB LOS classification, including a.o.: A decision tree‐based NLOS detection method for the UWB indoor location tracking accuracy improvement, Ardiansyah Musa  Gde Dharma Nugraha  Hyojeong Han  Deokjai Choi  Seongho Seo  Juseok Kim

At the end of the abstract (as well as in the conclusion of the paper), it is mentioned that the proposed solution is capable of mitigating ranging errors for LOS situations. This conclusion is not clear to the reviewer, as looking at the results in Figure 7, the results shows the highest improvement for NLOS situations, while LOS has little to no significant improvement. The reasoning behind this conclusion needs to be further explained.

The methodology of the proposed solutions is well-written and the general idea is clear. Nonetheless, the implementation details and chosen parameters of the machine learning models are not provided making result replication impossible. Additionally, a flowchart diagram would help clarify the flow of data to classification e.g. raw data (RSS )--> feature extraction (including which) --> model with parameter details --> classification / regression output. This could replace or complete Figure 1. The author would also like a clarification on why SOTA neural network techniques were not considered.

The related work seems thoroughly investigated, but the representation can be improved using a table. This table can clearly indicate other paper's findings / approaches / complexity requirements and accuracy results.

The result on a different MAE's considering LOS in both ternary and binary classification does not make sense. These should give the same results, as only the NLOS class changes (it becomes 1 or 2 classes). The only reasonable explanation for this is if the output NLOS/LOS classifier if directly used and different models, for each class that needs to mitigated, are trained. This should be better clarified and additionally, results of error mitigation using ground truth NLOS/LOS labels should be performed to optimise and evaluate only this mitigation sub-process.

Finally, the authors should clarify their vision on future work. While now being left in the dark, possible future approches can lead future researchers in the right direction when considering the conclusion that this approach cannot easily be generalized.

Reviewer 2 Report

The manuscript presents NLOS identification and mitigation techniques for UWB systems based on machine learning algorithms. The authors propose a classification of the link in two classes, LOS and NLOS, as well as three classes by splitting NLOS into NLOS-hard and NLOS-soft. Then regression is used to perform mitigation.

- Technical content and novelty.

The proposed analysis is interesting and deserves attention. NLOS propagation is one of the major impairments in UWB ranging and any attempt to mitigate in an effective way its effects may represent a valuable contribution. I do not see any serious flaw in the manuscript except a few aspects which I think the authors should better explain for the reader:

a) As far as I understand, the authors consider only RSSI as the feature to identify LOS and NLOS propagation. In doing this, authors completely discard the channel impulse response, which is very precious and detailed information about the channel in a UWB system. Isn't it a bit odd to perform classification discarding all such information?

b) When you mention RSSI, do you mean the received power from all the paths or only the power of the largest/first/(any other adjective) path? If I remember, DW devices can provide different metrics, one, for example, is the sum of the power of the first three paths. This aspect has to be also specified to help understand why RSSI is a reliable metric to deserve more attention than CIR.

c) Based on the conclusion given on Page 3, why remote ranging is a constraint? Isn't it an advantage? Doesn't it provide more flexibility? The justification given by the authors did not convince me satisfactorily.

d) I like NLOS-soft and -hard distinction. To me it seems a new point of view. However, the approach presented does not seem to perform very well in the three class classification. Maybe, again, CIR would be more informative to solve this problem as the physical meaning of NLOS-soft and NLOS-hard is very much related to the CIR.

e) The literature review could be improved a bit to make the state-of-the-art more comprehensive. For example, NLOS issues in UWB-based localization have been reviewed in "Ranging with ultrawide bandwidth signals in multipath environments," Proceedings of the IEEE, vol. 97, no. 2, pp. 404–426, Feb. 2009; "LOS/NLOS detection for UWB signals: A comparative study using experimental data"; "Position error bound for UWB localization in dense cluttered environments" IEEE Transactions on Aerospace and Electronic Systems. Another aspect that could be highlighted is the possible use of the author's idea, if this is hypothetically possible, to localization based on UWB radars such as in the systems presented in "Blind Selection of Representative Observations for Sensor Radar Networks," IEEE Transactions on Vehicular Technology, Apr. 2015 and "Sensor Radar for Object Tracking," Proceedings of the IEEE, June 2018. 

- Presentation.

The manuscript is generally well written and organized. There are some typos like, e.g.,   

"...they use the well know DW100" -> "well known", "and a target are classified " -> "is classified"

which need to be fixed. Overall the language can be improved and made more fluent after careful polishing.

Section 3.3 on Bayesian Optimization is rather short and not very informative. Please consider to expand it a bit more to make the contribution a bit more comprehensive.
